# Robust Conformal Prediction for Infrequent Classes

**Jens-Michalis Papaioannou**[*]  michalis.papaioannou@bht-berlin.de
*Berlin University of Applied Sciences and Technology*
*Leibniz University Hannover*

**Sebastian Jäger**[*]  sebastian.jaeger@bht-berlin.de
*Berlin University of Applied Sciences and Technology*

**Alexei Figueroa**  alexei.figueroarosero@bht-berlin.de
*Berlin University of Applied Sciences and Technology*
*Leibniz University Hannover*

**David Stutz**  dstutz@mpi-inf.mpg.de
*Max Planck Institute for Informatics*

**Betty van Aken**[†]  betty.vanaken@grammarly.com
*Grammarly*

**Keno Bressem**  keno.bressem@tum.de
*Technical University of Munich, School of Medicine and Health*

**Wolfgang Nejdl**  nejdl@l3s.de
*Leibniz University Hannover*

**Felix Gers**  FelixAlexander.Gers@bht-berlin.de
*Berlin University of Applied Sciences and Technology*

**Alexander Löser**  aloeser@bht-berlin.de
*Berlin University of Applied Sciences and Technology*

**Felix Biessmann**  felix.biessmann@bht-berlin.de
*Berlin University of Applied Sciences and Technology*
*Einstein Center Digital Future*

**Reviewed on OpenReview:** *https://openreview.net/forum?id=nJ4p8rh3Ig*

[*]Equal contribution.
[†]Work done while associated with Berlin University of Applied Sciences and Technology

## Abstract

Many real-world classification tasks involve datasets with large and imbalanced label spaces, making class-specific uncertainty quantification particularly challenging. Conformal Prediction (CP) provides a model-agnostic framework, which formally guarantees coverage, meaning that its prediction sets contain the true label with a user-defined probability (confidence level). However, standard class-conditional methods often fail when data is scarce for some classes. We propose a method that uses domain knowledge or label hierarchies to dynamically group semantically related classes to meet the desired coverage for a given confidence threshold. Our method maintains class-conditioned calibration when possible and provides group-conditioned guarantees where necessary. We evaluate our method on outcome diagnoses prediction, an important clinical task that does not only benefit from robust uncertainty estimation, but also presents a very imbalanced label distribution. We

conduct experiments using three clinical datasets employing two medical taxonomies (ICD-10 and CCSR) and label spaces of varying sizes with up to more than 1,000 classes. Our results show that the proposed approach is able to successfully exploit the label hierarchy and consistently improves class-conditional coverage for infrequent diagnoses. By improving coverage for underrepresented classes, our method enhances the reliability and trustworthiness of predictive models. This improvement is especially valuable in clinical applications, where failure to detect rare but serious conditions can lead to harmful consequences.

## 1 Introduction

In this work, we address class calibration in challenging settings with a large number of classes and limited available samples. We focus on tasks involving hierarchically organized label spaces, where classes are structured according to the relationships between the classes, e.g., a taxonomy. Such hierarchies capture semantic relationships between labels and are common in many real-world domains, including product categorization, biological classification of bacteria, or diagnoses in healthcare. More broadly, our approach applies to any setting where domain knowledge enables meaningful grouping of classes, whether through formal hierarchies or alternative semantic relationships. In settings where structured domain knowledge is limited, our method is complementary to data-driven approaches that automatically discover groupings from data.

We focus on the medical domain, specifically on the task of outcome diagnosis prediction as a canonical example of this setting. A key challenge of outcome diagnosis prediction is the large and imbalanced label space that exhibits a pronounced long-tail. Clinical decision support systems (CDSS) must not only show strong performance, but also be well-calibrated, as miscalibration can lead to harmful misdiagnoses (Alkan et al., 2025). At the same time, clinical models are usually designed to yield point predictions (Miotto et al., 2018; 2016), which offer no measure of uncertainty. This is especially problematic in diagnosis tasks, where overlapping symptoms (Wagan et al., 2024) are common and models may struggle to distinguish between similar conditions, especially for underrepresented classes. Thus, there is an important requirement for models to provide reliable predictions, as well as uncertainty estimates.

The medical domain is particularly suitable for demonstrating the impact of improvements in class conditional coverage, as it offers well-established taxonomies such as ICD-codes and presents high-stakes scenarios where calibration failures can directly affect patient safety. For a successful clinical deployment of AI technology, medical staff and patients need to trust its predictions. A central assumption underlying a substantial body of literature in *eXplainable AI* (XAI) is that trust can be fostered by rendering model predictions more transparent (Ribeiro et al., 2016; Lundberg & Lee, 2017; Samek et al., 2019; Schmidt & Biessmann, 2019) especially in the medical domain (Hamm et al., 2023; van Aken et al., 2022).

While transparency addresses an important dimension of trustworthiness (Wang & Yin, 2021), another key aspect lies in understanding the uncertainty of AI system predictions (Dhuliawala et al., 2023). In light of the growing capacity of models, which has been associated with poor calibration of uncertainty estimates (Snoek et al., 2019; Guo et al., 2017), improving uncertainty calibration is a fundamental prerequisite for trust in AI systems. This is especially true in safety-critical AI applications that fall under the high-risk category of the EU AI act (Council of European Union, 2024) such as AI healthcare products, where understanding uncertainty (Grote & Berens, 2023; Seoni et al., 2023) and communicating it (Banerji et al., 2023) are essential to improve trust.

**Uncertainty Calibration with Conformal Prediction.** A popular model-agnostic calibration method is Conformal Prediction (CP)(Vovk et al., 2005), which provides prediction sets instead of point predictions. These sets offer formal coverage guarantees, indicating how often the true label is expected to be included on average. This is especially important in clinical settings, where overconfident point predictions can be misleading. However, ensuring reliable coverage is challenging in imbalanced, long-tailed label distributions (Kasa & Taylor, 2023). This is because rare classes are often absent or severely underrepresented when calibration sets are very limited in size, making it difficult to estimate uncertainty reliably or guarantee valid coverage for those classes.

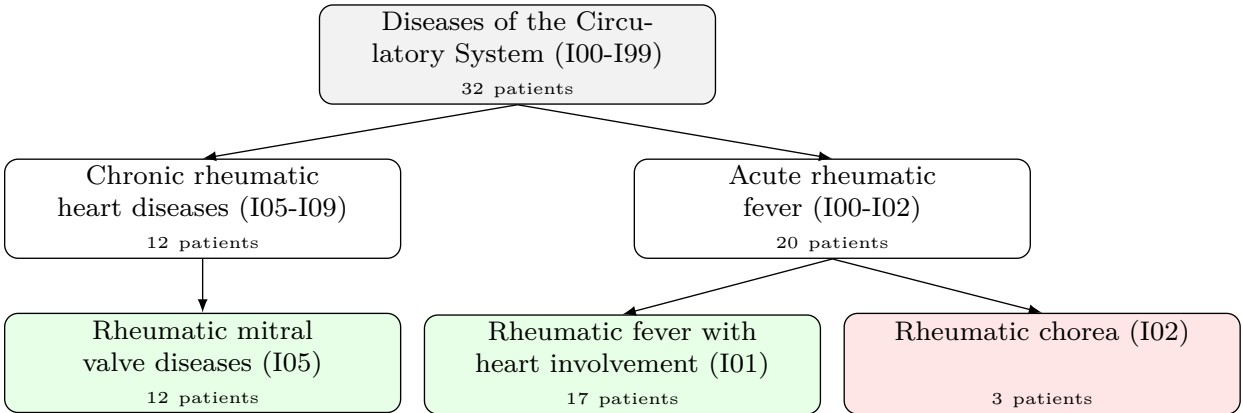

Figure 1: ICD-10 hierarchy of selected diseases of the circulatory system. ICD-10 codes are shown in parentheses and the number of patients in the calibration data per class is indicated at the bottom of each box. When the number of samples for a class is smaller than a certain threshold $m$ (here, $m = 10$), our method, *Dynamically Grouped Conformal Prediction (DGCP)*, groups this class with semantically related ones, using domain knowledge such as the ICD-10 hierarchy. In this example, the leaf node *Rheumatic chorea (I02)* with only 3 patients (highlighted in light red) is grouped with *I01* and *I05* because they share a common higher-level ICD category (indicated by gray shading), which is used to define the grouping. However, I01 and I05 exceed the threshold (highlighted in light green), which is why they are calibrated on the class level.

**Contribution: Dynamically Grouped Conformal Prediction.** We address these challenges by proposing a post-hoc and model-agnostic method called *dynamically grouped CP (DGCP)*. DGCP introduces a hyperparameter $m$, which defines the minimum number of calibration samples required for class-level calibration. This is motivated by preliminary experiments, which show that classes with no or only a few calibration samples cannot be reliably calibrated. Therefore, the idea is to relax the strict guarantees of class-conditional conformal prediction and dynamically group a class with fewer than $m$ samples together with semantically related classes using domain knowledge. As our experimental evaluation shows, this allows us to balance strong class-conditional guarantees, while increasing coverage for underrepresented classes. Figure 1 illustrates an example that uses ICD-10 codes as labels and $m = 10$. In this case, the number of calibration samples for the diagnosis of *rheumatic chorea (I02)* does not exceed the threshold $m$. Thus, DGCP combines all patients diagnosed with *I02* together with other diagnoses, using domain knowledge. For hierarchical label spaces, a natural grouping is given by a higher level in the hierarchy. However, it is also possible to use any other attribute or grouping function. Our experiments demonstrate that the proposed method is robust to the choice of hyperparameter $m$.

In summary, we propose *dynamically grouped conformal prediction* that maintains class-conditioned calibration if sufficient data are available and provides group-conditioned guarantees if not. We evaluate our approach on three clinical datasets and show that it consistently improves class-conditional coverage for the underrepresented classes.

## 2 Related Work

**Outcome Diagnoses Prediction from Text.** Transformer models have demonstrated remarkable performance across various domains, including the medical field. van Aken et al. (2021) pre-train transformers using a modified next-sentence prediction objective between admission and discharge sentences to improve outcome diagnoses prediction. Naik et al. (2022) augment clinical notes with medical literature and Ji & Marttinen (2023) adopts a multitask approach for unseen diagnoses categories. van Aken et al. (2022) addressed the problem of rare diagnoses codes by combining a prototypical classifier with a Transformer to improve prediction performance.

**Uncertainty Quantification and Conformal Prediction.** Uncertainty quantification in deep learning has gained considerable attention in recent years (Fakour et al., 2024; Tyralis & Papacharalampous, 2022; Abdar et al., 2021). Conformal prediction (CP) has emerged as a principled framework for producing prediction sets with rigorous coverage guarantees, even when the underlying models are imperfect. Notably, Straitouri & Rodriguez (2024); Straitouri et al. (2023) demonstrate that conformal prediction can assist domain experts reduce their workload, lead to better decisions, and increase trust (Dhuliawala et al., 2023).

Conformal prediction has been successfully applied across a wide range of domains, including natural language processing (Mohri & Hashimoto, 2024; Campos et al., 2024), clinical medicine (Hirsch & Goldberger, 2024; Grote & Berens, 2023; Banerji et al., 2023; Lu et al., 2022; Olsson et al., 2022; Vazquez & Facelli, 2022; Kompa et al., 2021), and drug discovery (Alvarsson et al., 2021), underscoring its broad utility. Further, a substantial amount of literature has focused on improving set efficiency (Dhillon et al., 2024; Stutz et al., 2022; Fisch et al., 2021; Romano et al., 2020; Angelopoulos et al., 2021), generalizing beyond coverage to other monotonic loss functions (Angelopoulos et al., 2024), tackling hierarchical classification (Mortier et al., 2025), distribution shifts (Gibbs & Candès, 2024; Barber et al., 2023; Bhatnagar et al., 2023) or structured output prediction (Zhang et al., 2025). In this work, we enhance the standard split conformal prediction framework (Angelopoulos & Bates, 2023) and propose an approach to improve class-conditional coverage for infrequent classes by incorporating domain knowledge.

The work most closely related to ours is by Ding et al. (2023), who address multiclass classification with up to 1,000 labels by clustering classes based on similar non-conformity scores. Like our work, their goal is to overcome the limitations of class-conditional conformal prediction in low-data regimes. Our approach is complementary and differs in two key aspects. First, while their method uses data-driven clustering to discover groupings from calibration data, we leverage domain knowledge such as label taxonomies to define semantically meaningful groups. This makes our approach particularly effective in domains where established hierarchical structures exist; However, data-driven methods are necessary when such knowledge is unavailable. Second, we apply grouping selectively: we maintain class-level calibration for well represented classes ($\geq m$ samples) and resort to group-level calibration only for underrepresented classes. This allows us to preserve fine-grained class-level guarantees where data are sufficient while ensuring valid coverage for rare classes through domain-informed grouping.

## 3 Task and Datasets

**Outcome Diagnoses Prediction.** We evaluate our approach on the task of predicting the primary discharge diagnosis from unstructured clinical admission notes, as introduced by van Aken et al. (2021). Unlike multi-label settings that consider multiple diagnoses per patient, this task focuses solely on the main diagnosis determined at discharge. Following van Aken et al. (2021), only information available at the time of admission is used for prediction, simulating a realistic early-decision support scenario. The task is formulated as a multiclass classification problem with up to over 1,000 possible labels, most of which are underrepresented in the training data.

**Datasets.** Large-scale, publicly available medical datasets for general use are rare. We use the MIMIC datasets, which are the most comprehensive clinical datasets that are publicly available. These contain anonymized patient records from the Intensive Care Unit (ICU) of the Beth Israel Deaconess Medical Center in Boston. MIMIC-III (Johnson et al., 2016) consists of data between 2001 and 2012, and MIMIC-IV (Johnson et al., 2023) between 2001 and 2019, respectively. To create datasets with label spaces of different sizes, we split MIMIC-IV randomly into halves. Each of the splits contains different patients. For the first dataset, we use three-digit ICD-10 codes (compare Choi et al. (2017)). We map the labels of the second MIMIC-IV dataset and MIMIC-III to *CCSR* codes, which are *clinically meaningful* groupings of ICD-10 codes (Healthcare Cost and Utilization Project (HCUP), 2024). We remove notes that directly mention the correct main diagnosis using MedCAT (Kraljevic et al., 2021). Additionally, we extract two attributes from each primary diagnosis. The first attribute, *body system*, is derived based on the classification framework provided by Healthcare Cost and Utilization Project (HCUP) (2024), which categorizes diagnoses into 21 clinically relevant groups, such as neoplasms (NEO), respiratory conditions (RSP), and injuries (INJ). Second, with the help of medical professionals, we assign a *severity score* per diagnosis, reflecting its level of life-threatening

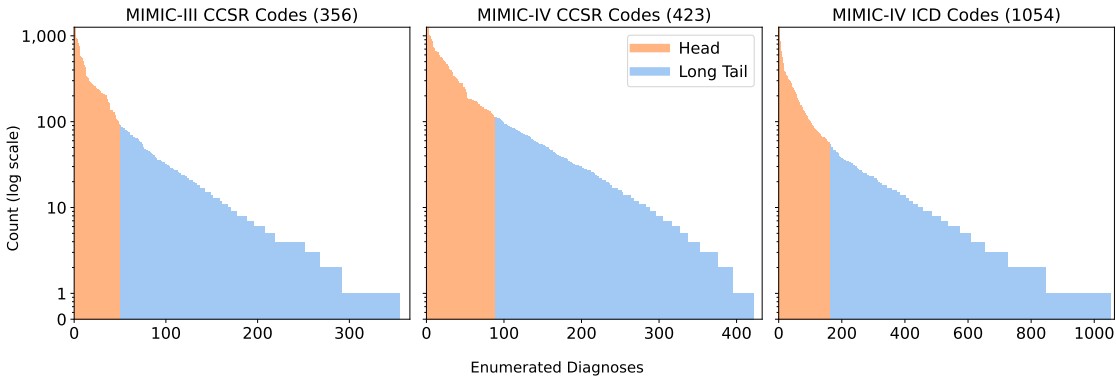

Figure 2: Diagnoses distribution in MIMIC-III (CCSR), MIMIC-IV (CCSR), and MIMIC-IV (ICD). Diagnoses are sorted by frequency within each dataset. The *Head* includes the most frequent diagnoses covering 80% of patients (e.g., 163 diagnoses in MIMIC-IV ICD), while the *Tail* comprises the remaining, less frequent diagnoses (e.g., 891 diagnoses in MIMIC-IV ICD) that account for the remaining 20% of cases. This highlights the extreme class imbalance present in clinical datasets.

risk. Severity level 1 corresponds to the most critical diagnoses such as sepsis. In contrast, severity level 5 represents non-critical diagnoses. This results in the following datasets: MIMIC-III (CCSR) with 356 diagnoses and $\approx 4.000$ records, MIMIC-IV (CCSR) with 423 diagnoses and $\approx 44.000$ records, and MIMIC-IV (ICD) with 1054 diagnoses and $\approx 44.000$ records. Note that the MIMIC-IV (CCSR) and MIMIC-IV (ICD) datasets not only differ in their label spaces, but also include different patients, resulting from a random split of MIMIC-IV in order to simulate two separate hospitals. In Figure 2 we present the label distribution of all training datasets.

We split each of the three datasets into training, validation, and test sets using stratified sampling. We keep each patient's first visit and ensure that all diagnoses appear at least once in the training set.

## 4 Background: Conformal Prediction

Conformal prediction (CP), originally presented by Vovk et al. (2005), is a distribution-free and model-agnostic uncertainty quantification method. It turns any black-box point predictor into a set predictor, which statistically guarantees to cover the correct label with a user-defined probability/confidence level. Assume $\hat{f}$ is a fitted classification model that outputs softmax scores: $\hat{f}(x) \in \mathbb{R}^K_{[0,1]}$. For point predictions, the predicted class $\hat{y} \in \{1, ..., K\}$ is the index of the highest softmax score.

**Non-conformity.** To build confidence sets $C(X_{test})$, CP uses non-conformity scores, computed on a calibration set not used for training: $S_{calib} = \{s_i\}, \forall i \in \{1, ..., N_{calib}\}$. Non-conformity scores $s_i$ represent how $(X_i, y_i)$ differs from the model prediction $(X_i, \hat{y}_i)$. For this, we use a non-conformity score function, e.g., one minus the softmax output of the true class: $s_i = 1 - \hat{f}(X_i)_{Y_i}$. Next, compute the $k$-th empirical quantile of $S_{calib}$ as follows:

$$k = \frac{\lceil (n+1)(1-\alpha) \rceil}{n}$$
$$\hat{q} = quantile(S_{calib}, k), \tag{1}$$

where $(1 - \alpha)$ is the user-defined confidence level and $n$ is the number of calibration points. For a new unseen test data point $X_{test}$ ($y_{test}$ is unknown), CP includes all classes in $C$ for which $s_i$ does not exceed the threshold $\hat{q}$. Formally, $C(X_{test}) = \{y : \hat{f}(X_{test})_y \le \hat{q}\}$, which is guaranteed to satisfy (Equation (2)), independently of the model and the data distribution (Zeni et al., 2020; Angelopoulos & Bates, 2023).

$$\mathbb{P}(y_{test} \in C(X_{test})) \ge (1 - \alpha) \tag{2}$$

Figure 3: Comparison of marginal (left) and conditional (right) conformal prediction. Both examples represent a test set of 100 data points, where 90% of the true labels exist in the conformal prediction sets, visualized as light gray points. Black points represent samples for which their true label was not included in the prediction set. Although both methods achieve 90% overall coverage, marginal prediction distributes coverage unevenly across groups. Where the groups are typically defined by the target class, but can theoretically depend on any attribute. On the other hand, conditional prediction guarantees 90% coverage for each group individually. This figure is adapted from Angelopoulos & Bates (2023).

**(Marginal) Coverage.** This property, referred to as *marginal coverage* (Lei & Wasserman, 2014), ensures that approximately $(1-\alpha)\%$ of the test data points are correctly included in the prediction sets. When the model $\hat{f}$ effectively fits the data, these sets $C$ tend to be small. Conversely, if $\hat{f}$ does not fit the data well or $X_{test}$ is ambiguous, $C$ will be greater in size (Lei et al., 2013). As Figure 3 (left) shows, although marginal coverage gives statistical guarantees on average, it may neglect the existence of groups in the data. Where the groups are typically defined by the target class, but can theoretically depend on any attribute. In many cases, it is desirable to obtain the coverage guarantee of Equation (2) for each group, known as conditional coverage.

**Conditional Coverage.** As illustrated in Figure 3, the left side (marginal coverage) shows that coverage is achieved for Group 1, while Group 2 achieves no coverage at all. However, because marginal coverage guarantees are only on average across all samples, the overall 90% confidence level is satisfied. To achieve a more balanced coverage, the formulation in Equation (1) is modified to define multiple group-specific thresholds $\hat{q}^{(g)}$, each corresponding to a single group $g \in \mathcal{G}$. As a result, associated variables such as $n^{(g)}$, $k^{(g)}$, and $S_{\text{calib}}^{(g)}$ are also indexed by group. Finally, the confidence sets are constructed as follows:

$$C(X_{\text{test}}) = \{y : \hat{f}(X_{\text{test}}^{(g)})_y \leq \hat{q}^{(g)}\},$$

where $g$ denotes the group to which the test sample belongs. These sets $C$ satisfy the stronger group-conditional guarantee defined in Equation (3), ensuring that each group individually meets the target coverage level. Note that in contrast to marginal, the right side (conditional coverage) of Figure 3 demonstrates balanced coverage across both groups, each achieving 90% coverage.

$$\mathbb{P}(y_{test} \in C(X_{test})|X_{test}^{(g)}) \geq (1-\alpha), \quad \forall g \in \mathcal{G} \tag{3}$$

**Class-conditional coverage.** If the groups are defined by a label attribute as follows:

$$g = \{k\}, \quad \forall g \in \mathcal{G} \text{ and } k \in \{1, ..., K\}, \tag{4}$$

Equation (3) is referred to as class-conditional coverage[1]. In many applications with small and balanced label spaces, applying class-conditional CP has shown good results. For further details and proofs, we refer the reader to Angelopoulos & Bates (2023); Vovk et al. (2005).

## 5 Methods

**Dynamical Grouping with Limited Calibration Samples per Class.** Reliable class-conditional coverage is challenging on datasets with large and imbalanced label spaces. To address this issue, we propose

---

[1]Class-conditional conformal prediction is also known as mondrian conformal prediction (Vovk et al., 2005).

dynamically grouping classes with *insufficient* calibration data into domain-specific groups. To this end, we introduce a new hyperparameter $m$, which denotes the minimum number of calibration samples required for a class to be considered sufficiently represented. If $|X_{\text{calib}}^{(k)}| \geq m$, class $k$ is treated as having *sufficient* calibration data, and we apply class-conditional conformal prediction with $g = \{k\}$ (cf. Equation (4)). Conversely, if $|X_{\text{calib}}^{(k)}| < m$, class $k$ is grouped together with other classes, and we perform group-based calibration according to Equation (3). The grouping function is user-defined and, given the calibration set size and a specific class, returns the appropriate group (e.g., based on additional features used to organize the data). For instance, in hierarchical label spaces, it may return a coarser-level category or another attribute describing, for example, the body system or severity of a disease. A formal definition of this procedure is provided in Algorithm 1.

---

**Algorithm 1** Calibration with Dynamically Grouped Conformal Prediction (DGCP)

---

**Require:** fitted classification model $\hat{f}$, calibration set $X_{calib}$, confidence level $(1-\alpha)$, sufficient data threshold $m$, grouping function group($\cdot$), non-conformity score function nonconformity($\cdot$)

**Ensure:** fitted classification model $\hat{f}$ is calibrated $\tilde{f}$

    $S_{\text{calib}} \leftarrow \text{nonconformity}(X_{calib}, \hat{f})$

    **for** each class $k \in \{1, \ldots, K\}$ **do**

        **if** $|X_{\text{calib}}^{(k)}| \geq m$ **then**

            $\hat{q}^{(k)} \leftarrow \text{Quantile}\left(S_{\text{calib}}^{(k)}, \frac{\lceil (n^{(k)}+1)(1-\alpha) \rceil}{n^{(k)}}\right)$                            $\triangleright$ class-level calibration

        **else**

            $g \leftarrow \text{group}(X_{calib}, k)$

            $\hat{q}^{(k)} \leftarrow \text{Quantile}\left(S_{\text{calib}}^{(g)}, \frac{\lceil (n^{(g)}+1)(1-\alpha) \rceil}{n^{(g)}}\right)$                       $\triangleright$ group-level calibration

        **end if**

    **end for**

    **return** calibrated model $\tilde{f}$

---

**Similarity to other Conformal Prediction Methods.** In contrast to Ding et al. (2023) who first cluster the calibration data based on their non-conformity scores, our method directly uses domain knowledge to assign samples to each group. In the best case, where every class has enough data for a reliable calibration, our method is equivalent to class-conditional CP (cf. Equation (4)) and fulfills the predefined class-conditional guarantees. If a class has fewer than $m$ calibration points, we group samples with a common attribute to estimate a group-specific threshold. This threshold is then applied to the underrepresented class. For example, diagnoses related to cardiovascular conditions can be grouped to estimate a shared threshold, which is then used for classes with limited data of that group (e.g., the rare *Takotsubo cardiomyopathy* diagnosis). This approach enables the calibration of classes that are absent from the calibration set.

## 6 Experimental Setup

Through a comprehensive evaluation, we validate empirically whether our methodology improves class-conditional coverage. Our primary objective is to improve upon existing methods that provide formal class-conditional coverage guarantees. While we also compare to non-conformal baselines that achieve strong empirical performance, these methods provide no formal guarantees on class-conditional coverage. Their good performance in our experiments reflects favorable empirical conditions rather than theoretical assurances and may not hold under other scenarios. Therefore, our key comparisons focus on methods with formal guarantees. We train a domain-specific model (Section 6.1) on each dataset, and compare the performance (Section 6.4) of different calibration methods (Section 6.2 and Section 6.3). For calibration, we use the following hyperparameters: calibration set sizes $n \in \{1000, 2000\}$ and confidence levels $(1 - \alpha) \in \{0.8, 0.9\}$ and keep the underlying predictors fixed. In addition, for DGCP, we evaluate the effect of $m \in \{10, 20\}$. To ensure the robustness of our results, we repeat each experiment 50 times. In each repetition, we randomly re-sample the calibration set from the test set and use the remaining data points for testing: 4,094 for

MIMIC-III, 20,900 for MIMIC-IV(CCSR), and 20,722 MIMIC-IV(ICD)). We report the metrics detailed in Section 6.4.

## 6.1 Prediction Model

For our experiments, we use ProtoPatient (van Aken et al., 2022), a transformer-based architecture that has demonstrated strong performance in diagnosis prediction, especially for rare diagnosis codes. The model combines a biomedical transformer encoder (Gu et al., 2020) with a prototypical layer. This layer consists of one prototype vector and one attention vector per diagnosis. Each patient admission note is encoded and projected into a lower-dimensional space, where it is weighed by diagnosis-specific attention vectors to map the note to the latent metric space. The model then computes the softmax over the negative distances between the resulting representation $v_{pat}$ and each diagnosis prototype $u_{diag}$, using the Euclidean distance: $d = \|v_{pat} - u_{diag}\|_2$. During training, the model minimizes the binary cross-entropy (BCE) loss over all patients $L = \sum_{pat} \sum_{diag} BCE(\text{softmax}(-d), y_{pat,diag})$, where $y_{pat,diag} \in \{0,1\}$ is the ground-truth. This loss encourages the representation of each input to move closer to the prototype of the correct class and farther from those of the incorrect ones. Finally, for inference, the prediction corresponds to the diagnosis with the closest prototype.

We choose ProtoPatient for evaluation because of its strong performance on rare diagnosis codes, and because its distance-based classification provides a natural non-conformity score. In addition, ProtoPatient offers inherent interpretability and justification for predictions, which complements our goals of transparency and trustworthiness, critical factors in clinical decision support. For further findings and results, we refer to van Aken et al. (2022). However, DGCP is model-agnostic and can be applied post hoc to any base predictor.

## 6.2 Conformal Calibration Methods

For model calibration, we define the non-conformity score as the distance between the patient encoding and the prototype of the true class: $s_i = d_i = \hat{f}(X_i)_{Y_i}, \forall i \in \{1, ..., N_{calib}\}$ (cf. Section 6.1), where $\hat{f}$ is the fitted ProtoPatient model. We apply CP as described in Section 4. Although we acknowledge that set size efficiency is important for practical deployment, in this study our focus is on improving class-conditional coverage. This choice reflects priorities in high-stakes tasks such as medical diagnosis, where including the true diagnosis (coverage) is critical.

**Marginal and Class-conditional CP.** To calibrate with marginal or class-conditional CP, we proceed as described in Section 5. Marginal CP provides coverage guarantees averaged over all labels (cf. Equation (2)). Class-conditional CP provides coverage guarantees for each class, accounting for imbalances that marginal coverage neglects (cf. Equation (4)).

**Clustered CP.** Clustered conformal prediction (Ding et al., 2023) improves class-conditional coverage in settings with limited data. It clusters classes with similar non-conformity scores and calibrates at cluster-level.

**Dynamically Grouped CP (ours).** We apply DGCP as described in Section 5 and use the following naming scheme: *DGCP/Grouping Method*. For example, when we use the *body system* related to a diagnosis for grouping, we refer to this method: *DGCP/Body System*.

## 6.3 Non-conformal Calibration Methods

For completeness, in addition to the above CP methods, we use baselines that construct prediction sets from model outputs, but do not provide any coverage guarantees.

**Adaptive top-k.** A simple approach that returns set predictions draws inspiration from the top-$k$ classification metrics. Where $k$ is not fixed, but classes are included in the prediction set until the cumulative sum of probabilities exceeds the predefined confidence level $(1 - \alpha)$.

**Calibrated Adaptive top-k.** Empirical evidence suggests that modern neural networks are poorly calibrated (Guo et al., 2017). To account for this, we use temperature scaling (TS) to calibrate ProtoPatient's probabilities and then follow the same approach as Adaptive top-k to construct prediction sets. Temperature scaling is a simple method to calibrate point prediction models by introducing a single scalar parameter $T > 0$, which scales the logit values $z$ and the calibrated logit $\tilde{z}$ is defined as:

$$\tilde{z} = \max_k \frac{\exp(z_k/T)}{\sum_{j=1}^{K} \exp(z_j/T)}, \quad \forall k \in \{1, \dots, K\}. \tag{5}$$

Note that in contrast to CP methods that are applied post-hoc, fitting $T$ requires gradient computation. We follow Angelopoulos et al. (2021) for the implementation.

### 6.4 Conformal Metrics

In our analysis, we use common metrics to assess the validity of our methods. Macro coverage measures the mean of the average empirical coverage per class and weights them equally. Formally,

$$coverage_{macro} = \frac{1}{K} \sum_{k=1}^{K} \frac{1}{|X_{test}^{(k)}|} \sum_{i=1}^{|X_{test}^{(k)}|} \mathbb{1}\big\{k \in C(X_{test,i}^{(k)})\big\} \tag{6}$$

where $K$ is the number of classes under consideration, typically the entire data $X$ or a subset thereof, $|X_{test}^{(k)}|$ is the number of test samples available for class $k$, and $C(X_{test,i}^{(k)})$ is the prediction set of the $i$-th example of class $k$. Ideally, the empirical macro coverage reaches the specified confidence level. Additionally, we measure the mean prediction set size:

$$set\_size = \frac{1}{|X_{test}|} \sum_{i=1}^{|X_{test}|} |C(X_{test,i})| \tag{7}$$

Smaller prediction sets are preferable due to their increased efficiency.

## 7 Results

Although we trained and evaluated two variants of the ProtoPatient model, one with pre-initialization of the prototypical layer following van Aken et al. (2022) and one with random initialization, we elaborate on their differences in Appendix A. Since the pre-initialized version consistently yields stronger conformal calibration metrics across all methods, we use it as the basis for all subsequent experiments. As described in Section 6, we calibrate the model with different approaches and compare them experimentally. If not stated otherwise, we show the results for $m = 10$, confidence level $(1 - \alpha) = 0.9$, and a calibration set size of 1000. We expand on the results of the different calibration hyperparameter settings in Appendix B.

Table 1 compares the performance of the calibration methods. We report the macro coverage over all classes in the dataset, as defined in Equation (6), which assigns equal weight to each class, including those lacking calibration or test samples, and the average prediction set size, as defined in Equation (7). Because all datasets exhibit pronounced long-tailed distributions, methods that neglect rare classes achieve lower macro coverage but smaller prediction set sizes. In this work, we primarily focus on improving class-conditional coverage. Enhancing the efficiency of prediction sets is left for future work.

Adaptive top-k and Calibrated Adaptive top-k show surprisingly good results and are generally second and third best in terms of macro coverage, but do not provide formal coverage guarantees. Clustered CP and Marginal CP perform almost equally in both class-conditional coverage and prediction set size. Class-conditional CP achieves the lowest macro coverage, as many classes do not appear in the calibration set, preventing the prediction of these classes.

Table 1: Experiment Results. Calibrated using $m = 10$, confidence level $(1 - \alpha) = 0.9$, and calibration set size of 1000. Conformal calibration methods (with guarantees) are above, and non-conformal methods (without guarantees) are below the horizontal line. Macro coverage is computed over the entire dataset $X$ and accounts for all classes present in the data. The corresponding number of classes is provided below each dataset name. Our approach, DGCP with body system fallback, consistently improves and outperforms other methods in terms of class-conditional coverage on all datasets. $\pm$ represents standard deviation over 50 repetitions.

| | Macro Coverage (↑) | | | Prediction Set Size (↓) | | |
|---|---|---|---|---|---|---|
| **Method** | MIMIC-III (356 CCSR) | MIMIC-IV (423 CCSR) | MIMIC-IV (1,054 ICD) | MIMIC-III (356 CCSR) | MIMIC-IV (423 CCSR) | MIMIC-IV (1,054 ICD) |
| DGCP\|Severity Score (ours) | $60.2 \pm 2.9$ | $67.0 \pm 1.6$ | $64.0 \pm 2.4$ | $17.0 \pm 2.8$ | $18.4 \pm 2.0$ | $33.9 \pm 6.7$ |
| DGCP\|Body System (ours) | $\mathbf{65.5 \pm 3.2}$ | $\mathbf{71.8 \pm 2.4}$ | $\mathbf{67.0 \pm 2.5}$ | $33.2 \pm 8.6$ | $31.0 \pm 5.7$ | $67.8 \pm 14.4$ |
| Clustered CP | $55.2 \pm 2.3$ | $66.5 \pm 2.1$ | $61.7 \pm 2.8$ | $12.0 \pm 1.1$ | $\mathbf{16.9 \pm 1.9}$ | $27.8 \pm 4.4$ |
| Class-conditional CP | $34.9 \pm 1.5$ | $38.9 \pm 1.5$ | $23.7 \pm 0.9$ | $16.3 \pm 1.1$ | $20.3 \pm 1.9$ | $\mathbf{21.4 \pm 2.2}$ |
| Marginal CP | $55.1 \pm 2.1$ | $66.6 \pm 1.7$ | $61.7 \pm 2.5$ | $\mathbf{12.0 \pm 1.0}$ | $17.0 \pm 1.6$ | $27.9 \pm 3.9$ |
| Calibrated Adaptive top-k | $64.1 \pm 1.6$ | $70.2 \pm 1.5$ | $64.5 \pm 1.6$ | $17.3 \pm 1.0$ | $21.9 \pm 1.8$ | $34.6 \pm 3.0$ |
| Adaptive top-k | $62.1 \pm 0.9$ | $67.4 \pm 0.2$ | $60.9 \pm 0.3$ | $15.3 \pm 0.1$ | $18.3 \pm 0.0$ | $27.5 \pm 0.0$ |

**Dynamically Grouped CP Improves Class-conditional Coverage.** DGCP\|Body System is consistently the best method in terms of macro coverage, closely followed by DGCP\|Severity Score. Both methods improve their class-conditional coverage over Class-conditional CP by a factor between 1.7 and 2.9. However, this typically comes at the cost of increased prediction set sizes; in our case, they range between 0.9 and 3.1. This highlights the trade-off between efficiency (small prediction set sizes) and reliability (high macro coverage). In our experiments, this relationship is roughly linear.

**A Closer Look at the Label Distribution in the Calibration Dataset.** Across 50 repetitions, only $166 \pm 6.4$ of the 356 classes in MIMIC-III are present in the calibration dataset, with $15.5\% \pm 1.4$ containing ten or more calibration samples. For MIMIC-IV (CCSR), $255 \pm 6.2$ out of 423 classes are represented in the calibration data, with $13.8\% \pm 1.2$ having at least ten samples. In MIMIC-IV (ICD), $313 \pm 9.3$ out of 1,054 classes appear in the calibration data, of which $5.4\% \pm 0.7$ include ten or more samples. In this experiment, the calibration dataset contains 1,000 samples, which explains why larger label spaces reduce these fractions. In Table 2, we show the macro coverage separately computed for classes in the calibration data that have ten or more calibration points and less than ten calibration points. Note that here we focus on the calibration dataset and compute the macro coverage over subsets that do not include all classes existing in the entire dataset. Consequently, these results must be interpreted independently of Table 1. As expected, for classes with sufficient calibration data, all conformal calibration methods consistently exceed the expected coverage of 90%. In contrast, for classes with fewer than ten calibration samples, coverage degrades. However, compared to Class-conditional CP all methods achieve improvements.

**Class-conditional Coverage under Data Scarcity.** Figure 4 shows the class-conditional coverage as a function of class size in the calibration data for all methods that provide class-level guarantees. Class-conditional CP yields the lowest conditional coverage, reaching the target confidence level $(1 - \alpha) = 0.9$ only once approximately ten calibration samples per class are available. For classes with fewer than ten samples, all methods exhibit increased class-conditional macro coverage. As discussed above, when sufficient calibration data are available (here $m = 10$), DGCP is equivalent to Class-conditional CP. Since Clustered CP behaves almost identically to Marginal CP, it attains close to 100% coverage for frequent classes.

To complement the analysis in Figure 4 and to extend the findings reported in Table 1, we additionally examine in Appendix C how coverage varies with respect to the class distribution encountered at test time. This provides further insight into the behavior of the methods when applied in practice.

Table 2: Experiment Results separated for classes with at least ten calibration points ($|X_{calib}^{(k)}| \geq 10$) vs. less than ten ($|X_{calib}^{(k)}| < 10$). Macro coverage is computed relative to these subsets in the calibration set and, therefore, represents only a fraction of all classes. Calibrated using $m = 10$, confidence level $(1 - \alpha) = 0.9$, and calibration set size of 1000. For classes with sufficient calibration data ($|X_{calib}^{(k)}| \geq 10$), all conformal calibration methods consistently exceed the expected coverage of 90%. However, with less calibration data, all methods achieve improvements over Class-conditional CP. DGCP|Body System followed by DGCP|Body System achieve the best improvements.

| | **Macro Coverage** ($\uparrow$) for $|X_{calib}^{(k)}| \geq 10$ | | | **Macro Coverage** ($\uparrow$) for $|X_{calib}^{(k)}| < 10$ | | |
| **Method** | MIMIC-III (356 CCSR) | MIMIC-IV (423 CCSR) | MIMIC-IV (1,054 ICD) | MIMIC-III (356 CCSR) | MIMIC-IV (423 CCSR) | MIMIC-IV (1,054 ICD) |
|---|---|---|---|---|---|---|
| DGCP|Severity Score (ours) | $92.1 \pm 1.3$ | $92.8 \pm 1.4$ | $92.6 \pm 1.7$ | $77.0 \pm 3.1$ | $77.5 \pm 1.9$ | $80.6 \pm 1.7$ |
| DGCP|Body System (ours) | $92.1 \pm 1.3$ | $92.8 \pm 1.4$ | $92.6 \pm 1.7$ | $\mathbf{77.7 \pm 2.4}$ | $\mathbf{81.6 \pm 1.8}$ | $\mathbf{82.6 \pm 1.8}$ |
| Clustered CP | $\mathbf{95.9 \pm 0.9}$ | $\mathbf{96.2 \pm 0.7}$ | $\mathbf{98.2 \pm 0.5}$ | $72.2 \pm 2.4$ | $76.6 \pm 2.3$ | $79.3 \pm 2.2$ |
| Class-conditional CP | $92.1 \pm 1.3$ | $92.8 \pm 1.4$ | $92.6 \pm 1.7$ | $66.8 \pm 2.9$ | $65.5 \pm 2.2$ | $62.6 \pm 2.1$ |
| Marginal CP | $95.8 \pm 0.8$ | $96.2 \pm 0.5$ | $\mathbf{98.2 \pm 0.5}$ | $72.1 \pm 2.3$ | $76.8 \pm 1.8$ | $79.3 \pm 1.9$ |

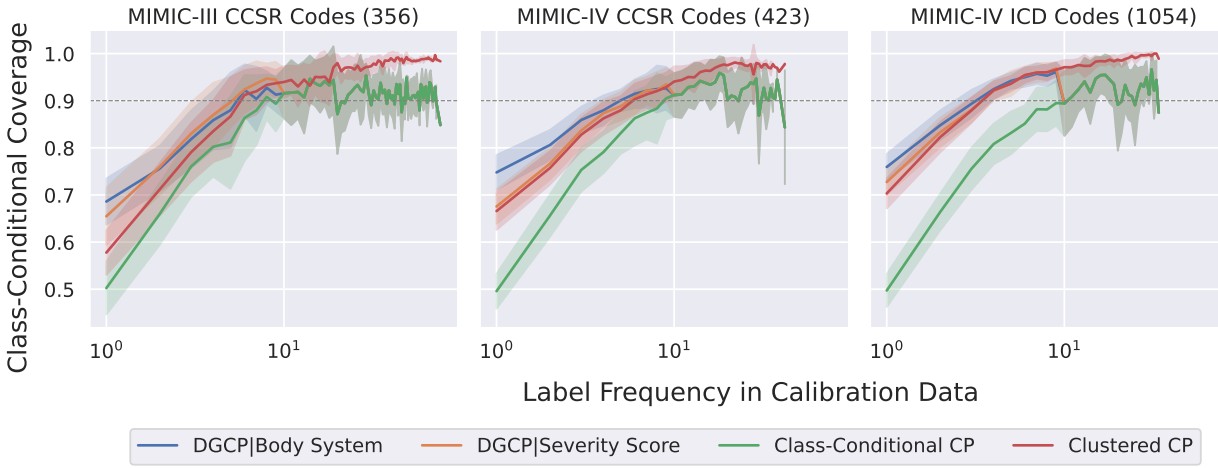

Figure 4: Class-conditional coverage as a function of class size in the calibration data for conformal methods. The horizontal dashed line marks the target confidence level. Calibrated using $m = 10$, confidence level $(1 - \alpha) = 0.9$, and a calibration set size $n = 1,000$. Error bars show the standard deviation over the 50 repetitions. If a class contains fewer than ten calibration samples, Class-conditional CP does not attain the target confidence level $(1 - \alpha) = 0.9$. In this regime, both DGCP variants and Clustered CP increase the class-conditional coverage. For classes with ten or more calibration samples, DGCP becomes equivalent to Class-conditional CP and reaches the target confidence level $(1 - \alpha) = 0.9$.

## 8  Discussion

**Non-conformal Calibration Methods.** Adaptive top-k and Calibrated Adaptive top-k achieve surprisingly strong macro coverage in our experiments, comparable to or exceeding some conformal methods. However, this empirical performance comes with an important caveat: these methods provide no formal guarantees. Under different conditions, their coverage may degrade arbitrarily, whereas conformal methods maintain their guarantees.

**Marginal CP and Class-conditional CP.** Class-conditional CP yields the lowest class-conditional coverage because it calibrates each class independently and cannot generate predictions for classes that are

absent from the calibration set. Given the highly imbalanced distribution of diagnoses and a limited calibration set of 1,000 samples, many classes remain unrepresented, leading to significantly degraded macro coverage. In contrast, Marginal CP aggregates across all samples during calibration, weighting classes according to their frequency in the calibration set. Unlike Class-conditional CP, it produces prediction sets for all classes, improving macro coverage. However, this comes at the cost of class-level guarantees. Since Marginal CP focuses on the majority classes (i.e., the head of the label distribution), it achieves relatively small prediction set sizes, but provides poorer coverage for infrequent classes.

**Clustered CP is Almost Equivalent to Class-conditional CP.** In terms of macro coverage and average set size, in our experiments, Clustered CP's results are almost equivalent to Marginal CP. This occurs because of the high imbalanced label space and the limited calibration set size. For the Clustered CP all classes that cannot be confidently assigned to any learned cluster are assigned to a NULL cluster, which uses marginal CP. Given the pronounced long-tail distribution of diagnoses and limited calibration data, almost all of the classes are assigned to this NULL cluster. Consequently, the NULL cluster dominates the overall behavior, causing the method to effectively behave similar to Marginal CP, which simply groups all classes together. In contrast, our domain knowledge-based grouping forms semantically meaningful groups, avoiding the need for a generic NULL category and enabling better calibration for underrepresented classes.

**Using Domain Knowledge Improves Class-conditional Coverage.** As shown in Table 1, leveraging domain knowledge to group diagnoses by body system or severity score based on medical taxonomies improves class-conditional coverage. In Table 2 and Figure 4, it is shown that improvements stem from the underrepresented classes rather than the majority classes. These results suggest that hierarchical label structures, which capture meaningful semantic relationships, can make calibration more robust, particularly for underrepresented classes. We argue that although this approach trades fine-grained class-level guarantees for more stable group-level estimation, the resulting increase in conditional coverage makes it a worthwhile compromise in imbalanced settings. Although not all domains offer expert-defined hierarchies or may introduce noise that complicates grouping, it could benefit domains with taxonomic label structures, such as the biological, legal, or financial domains.

## 9 Conclusion

In this work, we introduce *dynamically grouped conformal prediction (DGCP)*. We empirically demonstrate that our approach improves class conditional coverage in settings with limited data availability. By leveraging domain knowledge to group underrepresented classes, DGCP enables more robust threshold estimation for rare classes while preserving class-level guarantees for well represented ones. We demonstrated its effectiveness across three clinical datasets and two different label spaces: MIMIC-III (CCSR), MIMIC-IV (CCSR) and MIMIC-IV (ICD), showing consistent improvements in conditional coverage.

**Limitations and Future Work.** Our investigation focuses mainly on improving class conditional coverage for infrequent classes since these are critical for high-stakes domains. However, other aspects of conformal prediction are also important for safe clinical deployment. These include optimizing prediction set efficiency (Romano et al., 2020; Angelopoulos et al., 2021; Stutz et al., 2022) or controlling for alternative metrics such as F1-score (Angelopoulos et al., 2024) among others. Future work should explore the integration of these ideas into our approach.

## 10 Acknowledgments

We thank the reviewers and the Action Editor for their constructive feedback, which improved this work. This work is funded by the Deutsche Forschungsgemeinschaft (DFG, German Research Foundation) Project-ID 528483508 - FIP 12, European Union under the grant project 101079894 (COMFORT - Improving Urologic Cancer Care with Artificial Intelligence Solutions), German Federal Ministry of Research, Technology and Space grant number 16SV8856, as well as by BMWe SOOFI, Grant Agreement 13IPC040D. The views

expressed are solely those of the authors and do not necessarily reflect those of the European Union or European Health and Digital Executive Agency (HaDEA); neither is responsible for them.

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

## A    Influence of Pre-initialization of Prototypical Layer on Calibration

van Aken et al. (2022) introduced ProtoPatient and explored two initialization strategies: one in which prototypes and attention vectors are pre-initialized, and another using random initialization. Table 4 reports calibration results for all methods using model predictions from both initialization strategies, as indicated in the *Init* column. The results demonstrate that pre-initializing the prototypical layer yields higher macro coverage, smaller average prediction set sizes, and lower standard deviations across 50 runs with different random seeds, indicating more stable and reliable calibration. As shown in Table 3, pre-initialization also improves macro-AUC, consistent with the findings of van Aken et al. (2022), although it results in slightly lower accuracy. We argue that models with higher macro-AUC achieve better calibration results than those with higher accuracy.

Table 3: Performance Metrics of ProtoPatient on all datasets for the main clinical outcome prediction task. While AUC score consistently increases when pre-initializing is used, accuracy slightly degrades.

| Init | Dataset | #Classes | Accuracy | Macro-AUC |
|------|---------|----------|----------|-----------|
| False | MIMIC-III (CCSR) | 356 | 47.81 | 93.24 |
| | MIMIC-IV (CCSR) | 423 | 45.36 | 95.22 |
| | MIMIC-IV (ICD) | 1054 | 48.39 | 93.82 |
| True | MIMIC-III (CCSR) | 356 | 45.39 | 94.83 |
| | MIMIC-IV (CCSR) | 423 | 42.82 | 95.75 |
| | MIMIC-IV (ICD) | 1054 | 43.43 | 95.41 |

Table 4: Calibration results for ProtoPatient models trained with different pre-initialization methods (Init), using a calibration set size of 1,000 and a confidence level $(1 - \alpha) = 0.9$ and $m = 10$. Each row reports macro coverage (higher is better) and prediction set size (lower is better) for three clinical classification tasks: MIMIC-III (CCSR), MIMIC-IV (CCSR), and MIMIC-IV (ICD). Conformal methods with formal coverage guarantees appear above the single horizontal line; non-conformal baselines are shown below. Results are shown as mean ± standard deviation over 50 repetitions. Pre-initialization consistently improves calibration performance, yielding higher coverage, smaller prediction sets, and reduced variance.

| | | Macro Coverage (↑) | | | Prediction Set Size (↓) | | |
|---|---|---|---|---|---|---|---|
| Init | Method | MIMIC-III (CCSR) | MIMIC-IV (CCSR) | MIMIC-IV (ICD) | MIMIC-III (CCSR) | MIMIC-IV (CCSR) | MIMIC-IV (ICD) |
| Yes | DGCP\|Severity Score (ours) | $60.2 \pm 2.9$ | $67.0 \pm 1.6$ | $64.0 \pm 2.4$ | $17.0 \pm 2.8$ | $18.4 \pm 2.0$ | $33.9 \pm 6.7$ |
| | DGCP\|Body System (ours) | $65.5 \pm 3.2$ | $71.8 \pm 2.4$ | $67.0 \pm 2.5$ | $33.2 \pm 8.6$ | $31.0 \pm 5.7$ | $67.8 \pm 14.4$ |
| | Clustered CP | $55.2 \pm 2.3$ | $66.5 \pm 2.1$ | $61.7 \pm 2.8$ | $12.0 \pm 1.1$ | $16.9 \pm 1.9$ | $27.8 \pm 4.4$ |
| | Class-conditional CP | $34.9 \pm 1.5$ | $38.9 \pm 1.5$ | $23.7 \pm 0.9$ | $16.3 \pm 1.1$ | $20.3 \pm 1.9$ | $21.4 \pm 2.2$ |
| | Marginal CP | $55.1 \pm 2.1$ | $66.6 \pm 1.7$ | $61.7 \pm 2.5$ | $12.0 \pm 1.0$ | $17.0 \pm 1.6$ | $27.9 \pm 3.9$ |
| | Calibrated Adaptive top-k | $64.1 \pm 1.6$ | $70.2 \pm 1.5$ | $64.5 \pm 1.6$ | $17.3 \pm 1.0$ | $21.9 \pm 1.8$ | $34.6 \pm 3.0$ |
| | Adaptive top-k | $62.1 \pm 0.9$ | $67.4 \pm 0.2$ | $60.9 \pm 0.3$ | $15.3 \pm 0.1$ | $18.3 \pm 0.0$ | $27.5 \pm 0.0$ |
| No | DGCP\|Severity Score (ours) | $53.2 \pm 3.0$ | $66.6 \pm 2.9$ | $62.8 \pm 2.5$ | $20.0 \pm 2.6$ | $19.3 \pm 2.7$ | $38.6 \pm 8.8$ |
| | DGCP\|Body System (ours) | $60.3 \pm 3.4$ | $71.7 \pm 2.7$ | $65.1 \pm 2.4$ | $38.8 \pm 8.3$ | $34.4 \pm 6.4$ | $85.1 \pm 21.9$ |
| | Clustered CP | $44.4 \pm 3.6$ | $65.3 \pm 4.0$ | $60.5 \pm 2.7$ | $14.1 \pm 1.7$ | $16.4 \pm 2.4$ | $29.9 \pm 4.1$ |
| | Class-conditional CP | $34.7 \pm 1.7$ | $38.9 \pm 1.4$ | $23.8 \pm 0.8$ | $18.7 \pm 1.3$ | $20.6 \pm 2.1$ | $23.9 \pm 2.4$ |
| | Marginal CP | $44.3 \pm 3.2$ | $65.3 \pm 3.3$ | $60.7 \pm 1.9$ | $14.1 \pm 1.4$ | $16.3 \pm 1.9$ | $30.2 \pm 3.1$ |
| | Adaptive top-k | $57.3 \pm 1.4$ | $66.7 \pm 1.6$ | $61.5 \pm 1.4$ | $18.6 \pm 0.9$ | $17.5 \pm 1.3$ | $34.9 \pm 2.9$ |
| | Calibrated Adaptive top-k | $51.0 \pm 0.7$ | $53.9 \pm 0.2$ | $47.8 \pm 0.3$ | $15.0 \pm 0.1$ | $9.9 \pm 0.0$ | $16.8 \pm 0.0$ |

# B  Effect of Conformal Prediction Hyperparameters on Calibration

**Influence of Calibration Threshold** $m$**.**  The parameter $m$ specifies the minimum number of calibration samples required for class-level calibration. If a class contains fewer than $m$ samples in the calibration set, we group it with semantically related classes present in the set for threshold estimation. As shown in Table 5, increasing $m$ from 10 to 20 has minimal impact on class-conditional coverage and prediction set size across all datasets. A threshold of 20 leads to more frequent use of dynamic grouping, since in a calibration set of 1,000 samples it becomes more likely that individual classes will not meet the sample requirement. These results suggest that dynamic grouping introduces almost no effect (within the standard deviation) on coverage and prediction set efficiency, demonstrating the robustness of our method to modest changes in $m$.

**Influence of Confidence Level** $(1 - \alpha)$**.**  In Table 6, we present calibration results at a confidence level $(1 - \alpha) = 0.8$. To investigate the effect of varying the confidence level, we compare these results with those presented in Table 5, which uses the same calibration set size of 1,000 samples but a higher confidence level $(1 - \alpha) = 0.9$. As expected, lowering the confidence level leads to consistently smaller prediction set sizes across all datasets and methods. However, this reduction comes at the cost of decreased macro coverage, reflecting the fundamental trade-off between precision and reliability in conformal prediction. Despite the drop in coverage, the relative ranking of methods remains consistent, suggesting that method robustness is preserved across confidence levels. These findings emphasize the importance of choosing an appropriate confidence level based on task requirements, whether minimizing prediction ambiguity or maximizing empirical coverage with guarantees.

**Influence of Calibration set Size** $n$**.**  We present calibration results for two calibration set sizes: 1,000 samples in Table 5 and 2,000 samples in Table 7. We analyze the impact of calibration set size and compare the results at a fixed confidence level $(1 - \alpha) = 0.9$. The most notable gain in class-conditional coverage is observed for Class-conditional CP which shows a substantial increase in coverage across all datasets. However, this improvement comes at the cost of significantly larger prediction sets. In contrast, DGCP variants (based on severity and body system) maintain comparable coverage while showing a reduction in prediction set size and a decrease in standard deviation across repetitions. This suggests that our method benefits from the increase in calibration data by becoming more efficient and stable without sacrificing reliability. For other baselines, including Marginal CP, Clustered CP, and non-conformal methods, both coverage and set size remain largely unchanged, indicating limited sensitivity to calibration size. Therefore, in this study, we focus our in-depth analysis on the scenario with only 1,000 calibration samples to better understand performance in terms of class-conditional coverage under more limited data conditions.

Table 5: Calibration results for different values of $m$. All models use the pre-initialized model, a confidence level $(1 - \alpha) = 0.9$ and a calibration set size $n = 1,000$. Results show that decreasing $m$ from 20 to 10 maintains stable performance in terms of macro coverage and prediction set size. Conformal methods with formal coverage guarantees are listed above the single horizontal line, while non-conformal baselines are shown below. $\pm$ indicates standard deviation over 50 repetitions.

| | | Macro Coverage ($\uparrow$) | | | Prediction Set Size ($\downarrow$) | | |
| --- | --- | --- | --- | --- | --- | --- | --- |
| $m$ | Method | MIMIC-III (CCSR) | MIMIC-IV (CCSR) | MIMIC-IV (ICD) | MIMIC-III (CCSR) | MIMIC-IV (CCSR) | MIMIC-IV (ICD) |
| 10 | DGCP\|Severity Score (ours) | $60.2 \pm 2.9$ | $67.0 \pm 1.6$ | $64.0 \pm 2.4$ | $17.0 \pm 2.8$ | $18.4 \pm 2.0$ | $33.9 \pm 6.7$ |
| | DGCP\|Body System (ours) | $65.5 \pm 3.2$ | $71.8 \pm 2.4$ | $67.0 \pm 2.5$ | $33.2 \pm 8.6$ | $31.0 \pm 5.7$ | $67.8 \pm 14.4$ |
| 20 | DGCP\|Severity Score (ours) | $60.3 \pm 2.9$ | $67.2 \pm 1.6$ | $64.0 \pm 2.4$ | $16.5 \pm 3.0$ | $18.1 \pm 1.9$ | $34.8 \pm 7.0$ |
| | DGCP\|Body System (ours) | $65.5 \pm 3.2$ | $71.9 \pm 2.4$ | $67.0 \pm 2.5$ | $33.2 \pm 8.7$ | $31.3 \pm 5.9$ | $68.6 \pm 14.5$ |
| | Clustered CP | $55.2 \pm 2.3$ | $66.5 \pm 2.1$ | $61.7 \pm 2.8$ | $12.0 \pm 1.1$ | $16.9 \pm 1.9$ | $27.8 \pm 4.4$ |
| | Class-conditional CP | $34.9 \pm 1.5$ | $38.9 \pm 1.5$ | $23.7 \pm 0.9$ | $16.3 \pm 1.1$ | $20.3 \pm 1.9$ | $21.4 \pm 2.2$ |
| | Marginal CP | $55.1 \pm 2.1$ | $66.6 \pm 1.7$ | $61.7 \pm 2.5$ | $12.0 \pm 1.0$ | $17.0 \pm 1.6$ | $27.9 \pm 3.9$ |
| | Adaptive top-k | $64.1 \pm 1.6$ | $70.2 \pm 1.5$ | $64.5 \pm 1.6$ | $17.3 \pm 1.0$ | $21.9 \pm 1.8$ | $34.6 \pm 3.0$ |
| | Calibrated Adaptive top-k | $62.1 \pm 0.9$ | $67.4 \pm 0.2$ | $60.9 \pm 0.3$ | $15.3 \pm 0.1$ | $18.3 \pm 0.0$ | $27.5 \pm 0.0$ |

Table 6: Calibration results for calibration set size $n = 1,000$ and a confidence level $(1 - \alpha) = 0.8$. Each row presents the macro coverage (higher is better) and prediction set size (lower is better) across three clinical classification tasks: MIMIC-III (CCSR), MIMIC-IV (CCSR), and MIMIC-IV (ICD). Conformal methods with formal coverage guarantees are listed above the single horizontal line, while non-conformal baselines are shown below. The threshold $m$ indicates the minimum number of samples required for class-level calibration. Reported values are mean $\pm$ standard deviation over 50 repetitions.

| | | Macro Coverage ($\uparrow$) | | | Prediction Set Size ($\downarrow$) | | |
| --- | --- | --- | --- | --- | --- | --- | --- |
| $m$ | Method | MIMIC-III (CCSR) | MIMIC-IV (CCSR) | MIMIC-IV (ICD) | MIMIC-III (CCSR) | MIMIC-IV (CCSR) | MIMIC-IV (ICD) |
| 10 | DGCP\|Severity Score (ours) | $42.5 \pm 2.0$ | $50.6 \pm 1.8$ | $46.1 \pm 2.2$ | $6.5 \pm 0.8$ | $7.3 \pm 0.8$ | $11.2 \pm 1.8$ |
| | DGCP\|Body System (ours) | $49.0 \pm 2.9$ | $56.5 \pm 2.3$ | $49.9 \pm 2.4$ | $14.5 \pm 4.4$ | $12.4 \pm 2.5$ | $26.1 \pm 8.7$ |
| 20 | DGCP\|Severity Score (ours) | $42.5 \pm 2.1$ | $50.9 \pm 1.8$ | $46.2 \pm 2.2$ | $6.2 \pm 0.7$ | $7.4 \pm 0.8$ | $11.7 \pm 2.0$ |
| | DGCP\|Body System (ours) | $48.9 \pm 2.9$ | $56.7 \pm 2.4$ | $50.0 \pm 2.4$ | $14.4 \pm 4.3$ | $12.7 \pm 2.5$ | $26.5 \pm 8.7$ |
| | Clustered CP | $37.7 \pm 1.5$ | $50.1 \pm 2.8$ | $43.9 \pm 2.6$ | $5.1 \pm 0.5$ | $7.3 \pm 1.0$ | $10.3 \pm 1.6$ |
| | Class-conditional CP | $34.0 \pm 1.5$ | $38.1 \pm 1.5$ | $23.5 \pm 0.9$ | $13.6 \pm 1.3$ | $17.0 \pm 1.7$ | $19.7 \pm 2.2$ |
| | Marginal CP | $37.6 \pm 1.4$ | $49.9 \pm 2.0$ | $43.9 \pm 1.7$ | $5.0 \pm 0.4$ | $7.2 \pm 0.7$ | $10.2 \pm 1.0$ |
| | Adaptive top-k | $51.1 \pm 1.3$ | $55.8 \pm 1.4$ | $48.7 \pm 1.5$ | $8.0 \pm 0.4$ | $9.7 \pm 0.7$ | $13.9 \pm 1.0$ |
| | Calibrated Adaptive top-k | $48.9 \pm 0.8$ | $52.9 \pm 0.2$ | $45.6 \pm 0.3$ | $7.3 \pm 0.0$ | $8.4 \pm 0.0$ | $11.6 \pm 0.0$ |

Table 7: Calibration results for calibration set size $n = 2,000$ and a confidence level $(1-\alpha) = 0.9$. Each row presents the macro coverage (higher is better) and prediction set size (lower is better) across three clinical classification tasks: MIMIC-III (CCSR), MIMIC-IV (CCSR), and MIMIC-IV (ICD). Conformal methods with formal coverage guarantees are listed above the single horizontal line, while non-conformal baselines are shown below. The threshold $m$ indicates the minimum number of samples required for class-level calibration. Reported values are mean $\pm$ standard deviation over 50 repetitions.

| | | Macro Coverage (↑) | | | Prediction Set Size (↓) | | |
|---|---|---|---|---|---|---|---|
| $m$ | Method | MIMIC-III (CCSR) | MIMIC-IV (CCSR) | MIMIC-IV (ICD) | MIMIC-III (CCSR) | MIMIC-IV (CCSR) | MIMIC-IV (ICD) |
| 10 | DGCP\|Severity Score (ours) | $62.3 \pm 2.2$ | $67.3 \pm 1.1$ | $63.7 \pm 1.8$ | $15.5 \pm 1.3$ | $18.9 \pm 1.4$ | $30.4 \pm 3.3$ |
| | DGCP\|Body System (ours) | $68.1 \pm 2.6$ | $71.8 \pm 1.9$ | $66.3 \pm 2.0$ | $28.1 \pm 4.3$ | $27.0 \pm 3.2$ | $51.7 \pm 9.3$ |
| 20 | DGCP\|Severity Score (ours) | $62.3 \pm 2.2$ | $67.2 \pm 1.1$ | $63.9 \pm 1.8$ | $14.6 \pm 1.2$ | $16.5 \pm 0.9$ | $30.5 \pm 3.4$ |
| | DGCP\|Body System (ours) | $68.0 \pm 2.6$ | $71.8 \pm 1.9$ | $66.4 \pm 2.0$ | $27.1 \pm 4.2$ | $25.1 \pm 2.8$ | $52.6 \pm 9.5$ |
| | Clustered CP | $58.3 \pm 2.5$ | $67.0 \pm 1.6$ | $61.7 \pm 2.4$ | $12.2 \pm 0.9$ | $15.2 \pm 1.7$ | $25.6 \pm 4.0$ |
| | Class-conditional CP | $47.8 \pm 2.0$ | $51.2 \pm 1.5$ | $34.0 \pm 1.2$ | $22.6 \pm 1.5$ | $28.9 \pm 2.2$ | $34.3 \pm 2.8$ |
| | Marginal CP | $57.9 \pm 2.1$ | $67.1 \pm 1.2$ | $62.1 \pm 2.0$ | $11.9 \pm 0.6$ | $17.2 \pm 1.0$ | $28.0 \pm 3.2$ |
| | Adaptive top-k | $64.5 \pm 1.6$ | $69.1 \pm 1.1$ | $63.6 \pm 1.3$ | $15.6 \pm 0.6$ | $20.4 \pm 1.1$ | $32.1 \pm 2.2$ |
| | Calibrated Adaptive top-k | $64.2 \pm 1.5$ | $67.6 \pm 0.3$ | $61.2 \pm 0.4$ | $15.3 \pm 0.2$ | $18.3 \pm 0.0$ | $27.5 \pm 0.1$ |

## C Analysis of Class-Conditional Methods at Inference Time

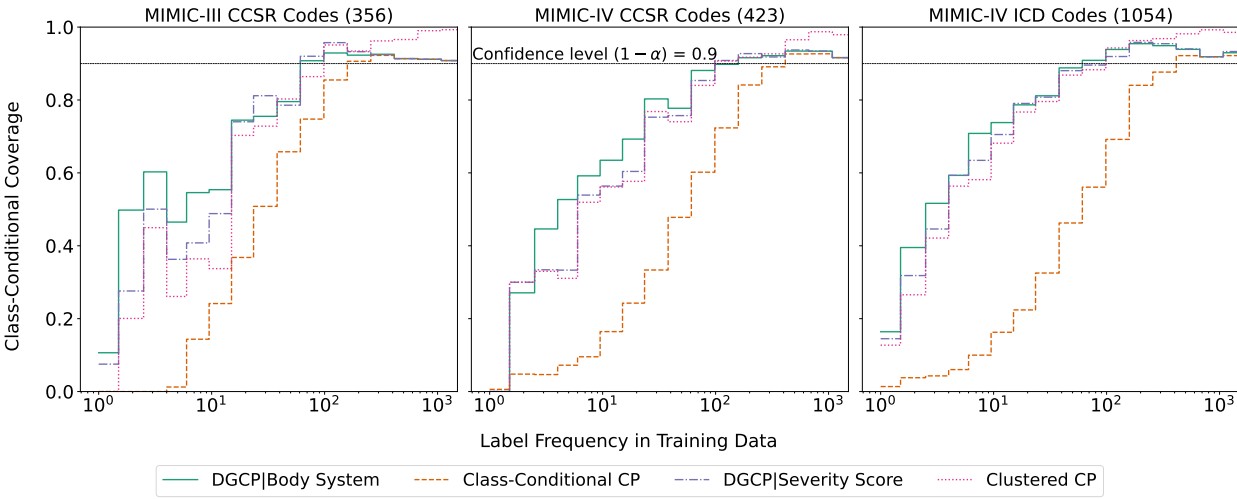

Figure 5: Class-conditional coverage as a function of training-set label frequency. The dashed horizontal line marks the target confidence level $(1 - \alpha) = 0.9$. Results shown for $(1 - \alpha) = 0.9$, calibration size $n = 1,000$, and threshold $m = 10$. Frequent classes typically have sufficient calibration support to reach the target coverage, while rare classes often lack enough calibration points. Grouping methods (DGCP|Body System, DGCP|Severity Score, Clustered CP) shift the coverage curves leftward, indicating more efficient use of limited calibration data and improved coverage for low-frequency classes.

To complement the analysis in Figure 4 and extend the findings reported in Table 1, we additionally examine how coverage behaves with respect to the true distribution of classes encountered at inference time. The figure illustrates the expected class-conditional coverage under the application distribution and shows patterns consistent with Figure 4, however, with generally lower absolute coverage values due to the increased difficulty of achieving class-level validity under the true long-tailed training distribution.

Moreover, even in datasets where the label space is smaller than the calibration set (e.g., MIMIC-III/IV CCSR), individual calibration splits may still contain fewer than ten examples for classes that are frequent in the training data. Thus, calibration-set frequency should not be interpreted as the true class frequency; it reflects only the calibration support available in a particular split. Coverage plotted against calibration frequency therefore captures the behaviour of a method under limited calibration data, whereas coverage plotted against training-set frequency reflects the expected reliability across head and tail classes at inference time, where grouping-based methods provide the greatest benefit due to the underlying long-tailed label distribution.

