# OpenReview forum: "Robust Conformal Prediction for Infrequent Classes"
_TMLR — Accepted by TMLR_

### Review · Reviewer_9sGw · 2025-09-03

**Summary Of Contributions:**

The manuscript proposes Dynamically Grouped Conformal Prediction (DGCP), a post-hoc calibration scheme for multi-class classification tasks with long-tailed, hierarchically organized label spaces. If a class has $<m$ calibration samples, DGCP merges it with its sibling classes from an existing taxonomy and calibrates on the group; otherwise, it reduces to class-conditional CP. Empirically, DGCP shows higher macro class-conditional coverage and reduced efficiency relative to marginal CP, class-conditional CP, clustered CP (Ding et al. 2023), and two top-k baselines.

**Strengths**
- The authors address an important practical gap and aim at improving coverage for rare classes in high-risk medical applications.
- The curated MIMIC III/IV data represents well the long-tailed classification setting considered here. Two different hierarchies (severity score, body system) are used in the experiments.

**Weaknesses**
- Macro coverage often falls short of the nominal 0.9 (even after grouping) on the test set, while the method's theoretical guarantees are emphasized up front. Additional metrics and analysis would help understand the results better (please see "suggested revisions").
- The method assumes that a semantic hierarchy is provided.
- Hyperparameter _m_ is fixed a priori; no data-driven or adaptive strategy is explored.
- Prediction sets for DGCP are 2–3× larger than baselines for the body system hierarchy.
- There are a few minor errors/inconsistencies and presentation issues throughout where additional details would strengthen the claim (please see "suggested revisions").

**Audience:**

Yes

**Audience Explanation:**

This approach would be of interest to those working on uncertainty quantification for risk-sensitive many-class classification settings. Rare classes / long tails are underexplored. Hierarchical label spaces are common beyond medicine (ImageNet, product taxonomies, biological taxonomies, legal codes).

**Claims And Evidence:**

No

**Claims Explanation:**

The central claim is that DGCP "consistently improves class-conditional coverage for infrequent [classes], outperforming strong baselines in all settings in terms of class-conditional coverage." While it makes sense to assume access to a hierarchy/grouping of classes informed by domain knowledge, the extra information does not seem to help much -- DGCP improves macro coverage only slightly, at the expense of hurting prediction efficiency quite significantly. It would be helpful to see a more in-depth analysis backed by additional metrics, stratified by head/tail groups, to better understand the results. I believe this may uncover opportunities to improve the results further.

**Requested Changes:**

While the empirical improvements are not very dramatic, the proposed approach in the specific setting considered is interesting. I would recommend acceptance provided that the following points are addressed. I've marked as "optional" suggestions that would simply strengthen the work in my view.

### Analysis
[Adaptive top-k baselines]
* There are several possible reasons adaptive top-k baselines show strong performance: (1) choice of the inverse quantile conformity score function (as opposed to the correct-class model probability that the authors used), (2) lack of the conformal calibration layer (does this step hurt somehow, as the calibration set is sparse for some classes?), and (3) $k$ chosen adaptively rather than determined a priori for all test samples, like $m$. It would be clarifying to disentangle these effects.
* Even though adaptive top-k was not wrapped in a conformal layer, I believe [a] should be cited. Given its strong performance, the inverse quantile conformity score of [a] may yield better performance than the score used by authors. Switching to this score for all the conformal methods considered in the experiments may enable "fairer comparisons," as all the baseline methods would be using the same "effective score" and we would be isolating the effect of (1). The current "marginal CP" would be replaced by [a].
* If adaptive top-k baselines are to be included without conformal calibration as is, it's fairer to include the calibration set into the training set of the classifier, so that they get access to the same amount of data.
* (Optional) Is there an explanation for why the "body system" grouping method leads to such a large increase in prediction set size?

```
[a] Romano, Sesia, and Candes. (2020). Classification with Valid and Adaptive Coverage. NeurIPS.
```

[Additional experimental detail]
- What is varied across the 50 repetitions? Initialization for the underlying predictor or just the train/calib/test splits?
- Eq 6 in Section 6.4 defines marginal coverage, but the text says that "We measure these metrics as macro-averages to highlight changes impacting all classes and not only the majority class." Please make the equations consistent with the metrics actually used.
- To understand why the coverage values reported in Table 1 are so low compared to the nominal level of 0.9, it would be helpful to supplement the macro average coverage metric with the worst-class coverage and/or report the head-only and tail-only macro average coverage metrics separately (using the head/tail definition in Fig 2). The idea is to get a better sense of the coverage distribution across head/tail groups; please also consider including a figure. The same goes for the efficiency metric, to understand why the prediction sizes are so large.
- Approximately what fraction of the test samples were assigned to the NULL cluster?
- How many points are in the test set? Is it possible the test set is too small for minority class evaluation?

### Methods
* The proposed approach assumes access to a hierarchy of classes. Could the authors please elaborate on what is meant by "possible to use any other variable or grouping function?" The text seems to suggest that all that's required is a way to group a given (rare) class to other semantically related classes -- in the absence of a full hierarchy, how can we construct a semantic grouping function for medical diagnosis applications?
* The following text is potentially misleading: "While [Clustered CP] relies on unsupervised clustering to group samples, we instead leverage semantic similarity of labels to dynamically aggregate samples only for underrepresented classes while using class-specific data for frequent classes." Clustered CP operates in settings with less information (i.e., where a hierarchy or semantic grouping function is not available).
* (Optional) As future extensions, could the authors discuss ways the hyperparameter $m$ can be determined in a data-driven way or adaptively per class depending on model performance?

### Minor errors and inconsistencies
* BNNs offer subjective uncertainty at best, and aren't "calibrated" in frequentist sense. So to state that "a broad spectrum of methods to calibrate uncertainty estimates" exists and only mention BNNs is awkward, especially as BNNs aren't used as baselines. Other non-conformal post-hoc calibration methods for multi-class classification include temperature/vector/matrix scaling (Guo et al. 2017 used alongside adaptive top-k), Dirichlet calibration (Kull et al. 2019)
* Conformity and non-conformity scores are mixed up throughout -- please correct the relevant definitions and equations
- Fixing citep and citet throughout

---

> ### Author Response · Authors · 2025-10-26
> **Response to Comments and Questions**
>
> We thank the reviewer for their valuable feedback.
>
> > - There are several possible reasons adaptive top-k baselines show strong performance: (1) choice of the inverse quantile conformity score function (as opposed to the correct-class model probability that the authors used), (2) lack of the conformal calibration layer (does this step hurt somehow, as the calibration set is sparse for some classes?), and (3) k chosen adaptively rather than determined a priori for all test samples, like m. It would be clarifying to disentangle these effects.
> > - Even though adaptive top-k was not wrapped in a conformal layer, I believe \[a\] should be cited. Given its strong performance, the inverse quantile conformity score of \[a\] may yield better performance than the score used by authors. Switching to this score for all the conformal methods considered in the experiments may enable "fairer comparisons," as all the baseline methods would be using the same "effective score" and we would be isolating the effect of (1). The current "marginal CP" would be replaced by \[a\].
>
> Thank you for raising this point. Since the focus of our investigation is on improving conditional coverage for underrepresented classes in high-stakes environments, we decided to empirically demonstrate that our approach can improve class-conditional coverage for the standard conformal prediction framework. Therefore, we consider as our main baselines primarily other conformal prediction methods that are able to provide formal guarantees.
>
> While we acknowledge that APS could potentially improve the performance of our methodology and that set efficiency is important for practical deployment, our choice to focus on coverage reflects the priorities in high-stakes environments, where including the true diagnosis (i.e., coverage) is critical, particularly for underrepresented classes.
>
> We also agree that the Adaptive top-k baseline is very similar to the APS conformity score, which could explain its good empirical performance despite the lack of formal guarantees. Nevertheless, we have added a citation of this work in the revised manuscript.
>
> > - If adaptive top-k baselines are to be included without conformal calibration as is, it's fairer to include the calibration set into the training set of the classifier, so that they get access to the same amount of data.
>
> Thank you for this suggestion. However, retraining 50 models per dataset to incorporate the 1,000 samples of the calibration set is computationally prohibitive. Moreover, since the calibration set represents only a small fraction of the data, we do not expect substantial changes in performance. This argument works in the opposite direction as well: if some classifiers were trained on more data, that could also explain performance differences.
>
> In summary, we believe that keeping a single model fixed and comparing post-hoc calibration methods, even though the top-k baselines skip the calibration step, represents a reasonable trade-off given the 50 repetitions.
>
> > - What is varied across the 50 repetitions? Initialization for the underlying predictor or just the train/calib/test splits?
>
> The predictor is trained once on the training data and remains fixed; only the calibration/test split changes. We have revised the manuscript to clarify this point.
>
> > - Eq 6 in Section 6.4 defines marginal coverage, but the text says that "We measure these metrics as macro-averages to highlight changes impacting all classes and not only the majority class." Please make the equations consistent with the metrics actually used.
>
> We agree that providing explicit definitions of the metrics is preferable and have updated the corresponding equations accordingly. Please note that, based on feedback from reviewer YHgb, the notation has been slightly modified.
>
> > - To understand why the coverage values reported in Table 1 are so low compared to the nominal level of 0.9, it would be helpful to supplement the macro average coverage metric with the worst-class coverage and/or report the head-only and tail-only macro average coverage metrics separately (using the head/tail definition in Fig 2). The idea is to get a better sense of the coverage distribution across head/tail groups; please also consider including a figure. The same goes for the efficiency metric, to understand why the prediction sizes are so large.
>
> Thank you for this remark. Please also see our response to reviewer ZwB8's comment regarding Figure 4. We have added the paragraph “A Closer Look at the Label Distribution in the Calibration Set” and Table 2, which provide details on the macro coverage for classes with 10 or more calibration data points, as well as for classes with fewer data points.

---

> > ### Author Response · Authors · 2025-10-26
> > **Responses Continued**
> >
> > > - Approximately what fraction of the test samples were assigned to the NULL cluster?
> >
> > Almost all classes are assigned to the NULL cluster. We have added the paragraph “Clustered CP is Almost Equivalent to Class-conditional CP” to the discussion, where we elaborate on this point.
> >
> > > - How many points are in the test set? Is it possible the test set is too small for minority class evaluation?
> >
> > The test set sizes are 4,094 for MIMIC-III, 20,900 for MIMIC-IV CCSR codes, and 20,722 for MIMIC-IV ICD (these numbers have been added to Section 6). While a single test set of this size might raise concerns about generalization, we expect to obtain reliable information about robustness by repeating the experiments 50 times.
> >
> > > - The proposed approach assumes access to a hierarchy of classes. Could the authors please elaborate on what is meant by "possible to use any other variable or grouping function?" The text seems to suggest that all that's required is a way to group a given (rare) class to other semantically related classes -- in the absence of a full hierarchy, how can we construct a semantic grouping function for medical diagnosis applications?
> >
> > Thank you for raising this point. We elaborate on it in several places throughout the manuscript, particularly in Section 5.
> >
> > > - The following text is potentially misleading: "While \[Clustered CP\] relies on unsupervised clustering to group samples, we instead leverage semantic similarity of labels to dynamically aggregate samples only for underrepresented classes while using class-specific data for frequent classes." Clustered CP operates in settings with less information (i.e., where a hierarchy or semantic grouping function is not available).
> >
> > Thank you for this important observation. We have revised the sentence to more accurately describe their approach.
> >
> > > - BNNs offer subjective uncertainty at best, and aren't "calibrated" in frequentist sense. So to state that "a broad spectrum of methods to calibrate uncertainty estimates" exists and only mention BNNs is awkward, especially as BNNs aren't used as baselines. Other non-conformal post-hoc calibration methods for multi-class classification include temperature/vector/matrix scaling (Guo et al. 2017 used alongside adaptive top-k), Dirichlet calibration (Kull et al. 2019)
> >
> > Thank you for pointing this out. We changed the text in the manuscript accordingly.
> >
> > > - Conformity and non-conformity scores are mixed up throughout -- please correct the relevant definitions and equations
> >
> > We have double-checked the manuscript but were unable to identify the issue. Could you kindly provide specific pointers so that we can address them?
> >
> > > - Fixing citep and citet throughout
> >
> > Thank you for pointing out these inconsistencies. We have carefully reviewed the manuscript and improved the citations.

---

### Review · Reviewer_YHgb · 2025-09-05

**Summary Of Contributions:**

- The paper proposes Dynamically Grouped Conformal Prediction (DGCP), a model-agnostic framework to improve conformal prediction for datasets that have large and imbalanced label spaces.
- The key idea is to retain class-conditional calibration when sufficient calibration samples exist, while dynamically grouping underrepresented classes using domain knowledge (e.g., ICD hierarchies, body systems, severity levels) when data are scarce. This approach improves class-conditional coverage for rare classes, an especially important property in clinical diagnosis prediction tasks.
- The method is evaluated on three clinical datasets and consistently outperforms other baselines in terms of coverage for infrequent classes.

**Audience:**

Yes

**Audience Explanation:**

TMLR’s audience includes researchers in uncertainty quantification and applied ML in safety-critical domains. Improving conformal prediction for infrequent classes is directly relevant to these communities, particularly in healthcare applications where rare but high-risk conditions matter. Beyond clinical settings, the idea of dynamically grouping classes based on hierarchies could be valuable in other domains with structured label spaces (e.g., biology, legal or product taxonomies). Thus, the findings would be of interest to both methodological researchers and practitioners

**Broader Impact Concerns:**

The work has positive societal implications: improving uncertainty estimates for rare conditions could strengthen clinical decision support and reduce missed critical diagnoses. Risks include over-reliance on potentially biased or incomplete taxonomies and the possibility that larger prediction sets may overwhelm clinicians. These concerns could be mitigated through careful grouping choices, human-in-the-loop evaluation, and attention to set-size efficiency.

**Claims And Evidence:**

No

**Claims Explanation:**

While the proposed method (DGCP) is evaluated on three clinical datasets with repeated experiments, the evidence presented is not sufficient to convincingly support the broad claims. The improvements in class-conditional coverage seem to have largely resulted from increased prediction set sizes rather than fundamentally better calibration, and this trade-off is not analyzed in depth. Furthermore, the experimental scale is limited to a few datasets, despite claims that the method is model- and data-agnostic and could generalize to other domains. Key methodological questions, such as how to handle domains without hierarchies or how to automatically identify semantically related classes, remain unanswered. Overall, the claims are only partially substantiated, and stronger experimental validation and methodological clarity are needed.

**Requested Changes:**

**Method**
- The technical contribution is relatively limited, as the main novelty lies in applying simple hierarchical grouping to conformal prediction.
- It is unclear how the method would operate in domains where no semantically related classes or expert-defined hierarchies exist.
  - While the authors note that some domains (biology, law, finance) offer hierarchical structures, the method’s reliance on such structures limits its generality.
- The paper does not address whether semantically related classes could be identified automatically, which would improve applicability beyond domains with curated taxonomies.

**Experiments**
- The experimental scale is relatively small: only three clinical datasets are used, all from MIMIC. Evaluations on broader domains or non-hierarchical datasets would make the claims of generalizability more convincing.
- The paper states that model- and data-agnostic guarantees suggest generalization to other datasets and architectures, but such evidence should have been included rather than deferred to future work.
- Repeating experiments 50 times is a strong point, but the improvements in class-conditional coverage may stem mainly from larger prediction set sizes rather than more effective calibration. A deeper analysis of this trade-off is needed.

**Writing**
- Some notations are not properly defined; for example, $S_{calib} = \{ \hat{f}(X_{calib})_k: k = y_{calib} \}$ is firstly presented as undefined.
- Citations are inconsistently formatted. For example, “The authors of (van Aken et al., 2021)” should use `\citet{}` instead of `\citep{}`.
- Similarly, phrasing like “Naik et al. (2022), augment” should be revised as "Naik et al. (2022) augment" to maintain grammatical consistency.

---

> ### Author Response · Authors · 2025-10-26
> **Responses to Comments and Questions**
>
> We thank the reviewer for their valuable feedback.
>
> > - It is unclear how the method would operate in domains where no semantically related classes or expert-defined hierarchies exist.
> >   - While the authors note that some domains (biology, law, finance)  offer hierarchical structures, the method’s reliance on such structures  limits its generality.
> > - The paper does not address whether semantically related classes  could be identified automatically, which would improve applicability  beyond domains with curated taxonomies.
>
> We position DGCP as complementary to unsupervised, data-driven methods such as Clustered CP. Specifically, DGCP enables the incorporation of domain knowledge when it is available. In domains where such knowledge is lacking, practitioners can instead rely on data-driven approaches like Clustered CP to define groups for calibration. We have clarified this distinction in multiple sections of the manuscript.
>
> > - The paper states that model- and data-agnostic guarantees suggest  generalization to other datasets and architectures, but such evidence  should have been included rather than deferred to future work.
>
> We agree with this comment and have removed the claim from the manuscript.
>
> > - Repeating experiments 50 times is a strong point, but the  improvements in class-conditional coverage may stem mainly from larger  prediction set sizes rather than more effective calibration. A deeper  analysis of this trade-off is needed.
>
> It is true that, depending on the grouping function used, DGCP increases the average prediction set size. Since the primary aim of this work is to improve coverage when insufficient calibration data is available, we report prediction set sizes to provide readers with insights into what they can expect when applying this method. For applications where set efficiency is critical, there exist complementary approaches that specifically focus on reducing prediction set sizes.
>
> > - Some notations are not properly defined; for example, $S\_{calib} = { \\hat{f}(X\_{calib})k: k = y{calib}}$ is firstly presented as undefined.
>
> We thank the reviewer for pointing out the inconsistent notations. We have reworked and aligned the notations in Section 6.1 and following with the definitions and conventions introduced in Section 4, ensuring greater consistency throughout the manuscript.
>
> > - Citations are inconsistently formatted. For example, “The authors of (van Aken et al., 2021)” should use \\citet{} instead of \\citep{}.
> > - Similarly, phrasing like “Naik et al. (2022), augment” should be  revised as "Naik et al. (2022) augment" to maintain grammatical  consistency.
>
> Thank you for pointing out these inconsistencies. We have reviewed the manuscript and improved the citations accordingly.

---

### Review · Reviewer_ZwB8 · 2025-10-12

**Summary Of Contributions:**

**Summary:**
This paper develops conformal prediction methods that achieve (approximate) class-conditional coverage. The main methodological idea is to use auxiliary domain information to group together semantically similar classes for which limited calibration data is available. Experiments on clinical diagnosis datasets show that this method improves class-conditional coverage on rare classes.

**Strengths:**
The main method is a new proposal that performs well empirically.

**Weaknesses:**
The overall novelty of the methodological contribution is small. Closely related techniques for obtaining (approximate) class-conditional coverage have been proposed in Ding et al. (2023). With that said, the proposed methods do offer some small empirical improvements over existing approaches. Additionally, I found some the specifics of the experimental set-up and results to be difficult to interpret (see below).

**Audience:**

Yes

**Audience Explanation:**

I believe the results of this paper would be of interest to readers working in conformal inference or other topics in uncertainty quantification.

**Broader Impact Concerns:**

I have no concerns on the broader ethical implications of this work.

**Claims And Evidence:**

Yes

**Claims Explanation:**

The main claim of the article is that DGCP provides better class-conditional coverage than existing approaches. This is supported empirically in Table 1 and Figure 4.

**Requested Changes:**

**Major Changes**
- I found the presentation of the empirical results and method to be somewhat imprecise. I believe the paper would benefit from a more explicit definition of the main method as well as the various quantities shown in Table 1 and Figure 4 (e.g. macro coverage, definition of the x-axis in Figure 4).
- I find the results of Figure 4 to be somewhat surprising. Most notably, if I am understanding the plot correctly, the authors find that for classes in which there are ~100 training samples available, class-conditional conformal prediction obtains only 60-80% coverage (versus a target coverage of 90%). This seems quite strange to me as at that sample size I would expect this method to deliver near-exact coverage.
- Related to the above, the text below Figure 4 suggests that much of the undercoverage is due to rare classes not being well represented in the training data. It would be helpful to have more specific quantitative information regarding the class frequencies and the sources of miscoverage. Additionally, it would be beneficial to have more information on the groupings output by DGCP in this experiment.

**Minor Changes**
- The definition of the conformal prediction set above equation (2) is stated with strict inequality. Typically, one would define this set to allow for equality (so it is $\hat{C}(X_{\text{test}}) =\\{y : \hat{f}(X_{\text{test}})_y \leq q\\}$). This can be important for classification tasks where the equality cases is needed to ensure coverage.
- In the equation display at the top of page 6 it would be helpful to explicitly notate that the group, g depends on y.
- What clustering method was used in the comparison to Ding et al. (2023)? Does the choice of clustering method effect the comparisons?
- Why were the top-k based methods excluded from figure 4?

---

> ### Author Response · Authors · 2025-10-26
> **Response to Comments and Questions**
>
> We thank the reviewer for their valuable feedback.
>
> > I found the presentation of the empirical results and method to be  somewhat imprecise. I believe the paper would benefit from a more  explicit definition of the main method as well as the various quantities  shown in Table 1 and Figure 4 (e.g. macro coverage, definition of the  x-axis in Figure 4).
>
> We agree with the reviewer’s comment and have extended the description of our method in Section 5, including a formal definition presented as Algorithm 1. Furthermore, we have refined the definition of macro coverage and provided a more detailed introduction to Table 1 and Figure 4.
>
> > I find the results of Figure 4 to be somewhat surprising. Most  notably, if I am understanding the plot correctly, the authors find that  for classes in which there are ~100 training samples available,  class-conditional conformal prediction obtains only 60-80% coverage  (versus a target coverage of 90%). This seems quite strange to me as at  that sample size I would expect this method to deliver near-exact  coverage.
>
> You correctly interpreted Figure 4, and your intuition is accurate. However, since the calibration dataset contains 1,000 data points and the number of classes ranges from over 300 to over 1,000, the combination of severe class imbalance, a large number of classes, and the limited calibration set size prevents the class-conditional CP from achieving the target confidence level of 90%.
>
> Thank you for pointing this out. We have added a new paragraph titled “A Closer Look at the Label Distribution in the Calibration Set” to elaborate on this issue.
>
> > Related to the above, the text below Figure 4 suggests that much of  the undercoverage is due to rare classes not being well represented in  the training data. It would be helpful to have more specific  quantitative information regarding the class frequencies and the sources  of miscoverage. Additionally, it would be beneficial to have more  information on the groupings output by DGCP in this experiment.
>
> Thank you for the observation. We agree that additional details were needed. Accordingly, we have added a new paragraph titled “A Closer Look at the Label Distribution in the Calibration Set”, which provides a more detailed explanation.
>
> > The definition of the conformal prediction set above equation (2) is stated with strict inequality. Typically, one would define this set to allow for equality (so it is $\hat{C}(X_{\text{test}}) =\{y : \hat{f}(X_{\text{test}})_y \le q q\}$). This can be important for classification tasks where the equality cases is needed to ensure coverage.
>
> Thank you for spotting this important detail. You are correct, it should allow equality rather than strict inequality, as correctly stated in the text. We have corrected the formulas above Equations (2) and (3).
>
> > In the equation display at the top of page 6 it would be helpful to explicitly notate that the group, g depends on y.
>
> It is not required that $g$ depends on $y$ in the general case. We have made a small modification to Equation (3) to account for this more general formulation.
>
> > What clustering method was used in the comparison to Ding et al. (2023)? Does the choice of clustering method effect the comparisons?
>
> For Clustered CP, we used Ding et al.’s weighted k-means algorithm on the conformity scores. While the choice of clustering algorithm may influence performance, we expect only modest differences across reasonable alternatives. The key distinction lies in how the groups are defined: data-driven clustering versus expert-knowledge-based taxonomies. However, the extreme class imbalance, where many classes have very few training samples, limits the effectiveness of data-driven approaches that rely on sufficient data to estimate conformity score distributions.
>
> > Why were the top-k based methods excluded from figure 4?
>
> In Figure 4, we focus on conformal methods that provide class-conditional guarantees only; therefore, we omit the top-k-based methods and marginal CP. The main purpose of this plot is to illustrate that class-conditional conformal methods eventually achieve $confidence\_level \ge confidence_level$. Including additional methods could distract from this point. We have slightly revised the caption to clarify that only class-conditional conformal methods are shown and that the omission of other methods is intentional.

---

> > ### Comment · Reviewer_ZwB8 · 2025-11-10
> > **Additional Clarification**
> >
> > Thank you for the clarifications. I think the given revision has improved the manuscript. I still have two points of confusion about the results.
> >
> > 1) The results in Table 2 appear to be inconsistent with the results in Table 1. For instance, in Table 2 on MIMIC 3 Class-conditional CP has a macro-coverage of 0.921 on classes with >= 10 calibration points and a macro-coverage of 0.668 on classes with < 10 calibration points. However, in Table 1 the macro-coverage on the combined set of all classes is 0.349. Based on the given definitions, it seems to me that the macro-coverage on all classes should be a convex combination of the macro-coverage on the two subsets. Thus, I'm not sure how these results were obtained.
> >
> > 2) Upon taking a second look at section 7, I believe my confusion about Figure 4 may stem from the fact that the x-axis of this figure gives the number of samples in the training set and *not* the number of samples in the calibration set. Thus, it is possible that there are classes for which there are many training samples, but no samples in the calibration set. If my understanding is correct, I think this should be clarified more explicitly in the paper. Additionally, I think it would be (perhaps more) informative to see how the coverage varies with the calibration sample size, not just the training sample size.

---

> > > ### Author Response · Authors · 2025-11-14
> > > **Response to 'Additional Clarification'**
> > >
> > > Thank you for getting back to us. We respond inline below.
> > >
> > > > 1. The results in Table 2 appear to be inconsistent with the results in Table 1. For instance, in Table 2 on MIMIC 3 Class-conditional CP has a macro-coverage of 0.921 on classes with >= 10 calibration points and a macro-coverage of 0.668 on classes with < 10 calibration points. However, in Table 1 the macro-coverage on the combined set of all classes is 0.349. Based on the given definitions, it seems to me that the macro-coverage on all classes should be a convex combination of the macro-coverage on the two subsets. Thus, I'm not sure how these results were obtained.
> > >
> > > Thank you for pointing out this ambiguity. The subsets in Table 2 are derived from the calibration dataset, which is itself a subset of all classes presented in Table 1. Consequently, the quantities reported in Table 2 are not directly comparable to those in Table 1 and should be interpreted independently. We have revised the captions and the text preceding Table 1 to clarify this and to prevent future confusion.
> > >
> > >
> > > > 2. Upon taking a second look at section 7, I believe my confusion about Figure 4 may stem from the fact that the x-axis of this figure gives the number of samples in the training set and not the number of samples in the calibration set. Thus, it is possible that there are classes for which there are many training samples, but no samples in the calibration set. If my understanding is correct, I think this should be clarified more explicitly in the paper. Additionally, I think it would be (perhaps more) informative to see how the coverage varies with the calibration sample size, not just the training sample size.
> > >
> > > We agree that the original version of Figure 4 may be misleading. We therefore replaced it with a revised figure, following the reviewer’s suggestion, that shows coverage as a function of the calibration class size. At the same time, we believe the original plot still provides useful insight, as it illustrates how the methods behave when applied in practice, where training-class frequencies are often more readily available. For this reason, we moved the original figure to Appendix C and expanded the accompanying discussion to clarify the conclusions drawn from both figures.

---

### Author Response · Authors · 2026-01-27
**Thank You to All Involved**

Thank you very much for accepting the paper and approving the camera-ready version. We are grateful to the reviewers for their valuable feedback and for the opportunity to improve the quality and clarity of the manuscript.

---

### Decision · Action_Editor_vSSM · 2026-01-02

**Recommendation:** Accept as is

**Audience:**

Yes

**Audience Explanation:**

The contributions of this paper are likely appeal to members of TMLR's audience who work on uncertainty quantification and applied machine learning in risk-sensitive domains. Two aspects of this work are particularly interesting. First, the proposed method targets rare classes, which is an underexplored topic in uncertainty quantification. Second, it takes advantage of hierarchically structured labels which are often present in healthcare applications as well as in biological and legal settings.

**Claims And Evidence:**

Yes

**Claims Explanation:**

This paper centers around two key claims. The first claim is the introduction of an approach to conformal prediction that dynamically groups classes to achieve target coverage. The second claim is that the proposed method makes use of a label hierarchy and improves class-conditional coverage for rare classes on a medical diagnosis prediction task across three datasets. The proposed algorithm is outlined in Section 5 and the results shown in Table 1 and Figure 4 support the empirical claim.

The reviewers noted that the paper initially claimed to suggest generalization of the method to other "datasets, domains, and model architectures." This sort of generalization was not supported by the experiments in the paper, which focused on three datasets for medical diagnosis prediction. During the rebuttal phase, the authors reduced the scope of the claims to match the evidence.

The TMLR acceptance criteria handles this case explicitly. It states that authors may resolve gaps between claims and evidence by simply reducing their claims. Although the claims of the paper are limited in scope, they are adequately supported by evidence.